# Synergistic plasticity of intrinsic conductance and electrical coupling restores synchrony in an intact motor network

Brian J Lane[1][†], Pranit Samarth[2][†], Joseph L Ransdell[1], Satish S Nair[2], David J Schulz[1]*

[1]Division of Biological Sciences, University of Missouri-Columbia, Columbia, United States; [2]Department of Electrical and Computer Engineering, University of Missouri-Columbia, Columbia, United States

**Abstract** Motor neurons of the crustacean cardiac ganglion generate virtually identical, synchronized output despite the fact that each neuron uses distinct conductance magnitudes. As a result of this variability, manipulations that target ionic conductances have distinct effects on neurons within the same ganglion, disrupting synchronized motor neuron output that is necessary for proper cardiac function. We hypothesized that robustness in network output is accomplished via plasticity that counters such destabilizing influences. By blocking high-threshold $K^+$ conductances in motor neurons within the ongoing cardiac network, we discovered that compensation both resynchronized the network and helped restore excitability. Using model findings to guide experimentation, we determined that compensatory increases of both $G_A$ and electrical coupling restored function in the network. This is one of the first direct demonstrations of the physiological regulation of coupling conductance in a compensatory context, and of synergistic plasticity across cell- and network-level mechanisms in the restoration of output.

*For correspondence: SchulzD@missouri.edu

[†]These authors contributed equally to this work

**Competing interests:** The authors declare that no competing interests exist.

## Introduction

The hallmarks of robust central pattern generator (CPG) output are appropriately tuned excitability of individual neurons combined with circuit-level interactions that maintain appropriate temporal coordination (i.e., phasing) of these neurons. Through both developmental and ongoing tuning processes, CPGs can maintain reliable network output for decades across the lifespan of an individual, despite constant feedback from the changing nature of both the organismal and natural environment. Yet underlying this constant reliability of network output exists a surprising amount of variability in the individual parameters necessary for producing activity. For instance, despite having nearly identical output across animals, networks can exhibit a five-fold or more range in intrinsic and synaptic conductance values (*Marder and Goaillard, 2006*; *Schulz et al., 2006*; *Marder, 2011*; *Roffman et al., 2012*). Such variability in intrinsic conductances is not limited to CPGs, but has been documented in several cell types of the mammalian brain, including cerebellar Purkinje cells (*Swensen and Bean, 2005*), and globus pallidus neurons (*Günay et al., 2008*). Additionally, the synaptic strengths between mammalian central neurons have been shown to vary in several brain regions (*Nelson and Turrigiano, 2008*; *Turrigiano, 2008*; *Maffei et al., 2012*).

The origins and implications of this variability are still an intense area of investigation (*Krubitzer and Kahn, 2003*; *Turrigiano and Nelson, 2004*; *Ciarleglio et al., 2015*). We hypothesize that this variability might be a result of ongoing compensatory changes required to maintain reliable

**eLife digest** Neurons can communicate with each other by releasing chemicals called neurotransmitters, or by forming direct connections with each other known as gap junctions. These direct connections allow electrical impulses to flow from one neuron to another via pores in the membranes between the cells. Unlike communication via neurotransmitters, gap junctions are usually thought to be hard-wired and unchanging over the life of the animal.

Lane et al. recorded electrical activity in a network of neurons that generates rhythmic heart contractions in the Jonah crab. Neurons in this network usually all fire an electrical impulse at the same time, which is crucial to make sure that the whole heart contracts at the same time. The experiments show that drugs that block potassium channel pores in the membrane cause the neurons to fire too much and at different times to each other.

However, the network of neurons soon adapted to the changes caused by the drugs and returned to working as normal. Mimicking these changes in a computer model of the neuron network, together with experimental data, showed that changes to the gap junctions play a major role in restoring normal activity to the network.

The next step following on from this research is to understand how a network of neurons 'senses' that it is not working normally and changes its electrical activity.

output over time. This compensation, termed homeostatic plasticity, has been well documented both for plasticity of intrinsic excitability via changing ionic conductances (*Turrigiano et al., 1994*), as well as for changes in chemical synaptic strength (*Desai, 2004*; *Turrigiano, 2012*). Variability in conductances may also be an adaptive trait in and of itself: variable solutions that produce convergent circuit output may provide a selective advantage, or perhaps be a substrate for adaptation and evolution (*Marder and Goaillard, 2006*; *Grashow et al., 2009*).

Regardless of whether such variability is the result of homeostatic compensation, differential tuning across networks, or a combination of these and other heretofore undiscovered causes, a potential cost to such variability has recently been identified. In the cardiac ganglion (CG) of the Jonah crab (*Cancer borealis*), five Large Cell motor neurons (LCs) generate completely synchronous output, as a result of pacemaker inputs within the network, to drive simultaneous heart muscle contraction in the crab (*Tazaki, 1972*). Despite completely uniform and synchronous activity within the network, LCs show highly variable underlying maximal conductances (*Ransdell et al., 2012*). These variable conductances render the neurons susceptible to perturbations that target a subset of ionic conductances: when high-threshold $K^+$ currents were blocked with tetraethylammonium (TEA), the motor neurons lost coordinated output and became divergent in their patterns of firing (*Ransdell et al., 2013a*). These CG neurons compensate for this change in excitability, presumably to homeostatically maintain a target level of excitability (*Ransdell et al., 2012*). However, none of the network level impacts of this perturbation and plasticity have been investigated. Indeed, it is difficult to study homeostatic plasticity in intact networks and to simultaneously take into account both properties of individual cells as well as their network interactions. In the present study, we discovered that LC variability makes the network vulnerable to desynchronization as a result of TEA exposure, but that compensation resynchronizes the network within 30–60 min via both intrinsic cellular and circuit-level physiological mechanisms. To examine the underlying mechanisms, we developed a biophysical computational model of the entire cardiac network. The network model enabled a comprehensive search of the conductance space for potential compensatory mechanisms that preserved network synchrony, and we used these findings to guide further experimentation. Our study revealed cooperative homeostatic plasticity among intrinsic conductances and electrical coupling across multiple cells in the cardiac network. We interpret this as a novel homeostatic compensatory mechanism contributing to the overall robustness of CPG output.

# Results

## Exposure to TEA desynchronizes LC burst waveforms and increases excitability

*Ransdell et al. (2013a)* were able to repeatedly reduce the magnitude of high-threshold $K^+$ currents ($I_{Kd} + I_{BKKCA}$) by ~92% in isolated LCs with 25 mM TEA. We used this experimental manipulation on the 3 anterior LCs in intact CGs (*Figure 1A*) causing LCs in the same ganglion to change from identical (*Figure 1B*) to divergent, asynchronous output (*Figure 2A,I,II*) after exposure to TEA. After application of TEA, motor neurons became noticeably more depolarized during burst potentials and LC spiking seen on the extracellular recordings increased substantially (*Figure 2A*). Additionally, comparison of intracellular voltage waveforms revealed a loss of conserved output (*Figure 2A*). These results are consistent with the hypothesis that variable underlying conductances of the LCs makes them vulnerable to a uniform perturbation of a subset of conductances such as the TEA blockade that targets high-threshold $K^+$ conductances. Because our previous results demonstrated that the change in excitability that accompanies TEA exposure in LCs is accompanied by an increase in A-type $K^+$ current, we hypothesized that compensation may also occur at the network level to restore synchrony among LCs subsequent to the TEA block.

## Compensation restores both excitability and synchrony following TEA exposure

To determine whether compensatory responses can restore both excitability and synchrony of LC output following TEA block, anterior LCs were superfused with TEA for at least 1 hr while the activity of the individual LCs and the network were continuously recorded. Our data demonstrate that both synchrony and excitability are restored towards baseline levels over a period of 30–60 min following TEA exposure. *Figure 2A and B* illustrate a typical progression through the loss and subsequent restoration of synchrony among LCs during one hour of continuous exposure to TEA. Acutely after application of TEA, we saw a significant reduction in synchrony as measured by $R^2$ (see Materials and methods; *Ransdell et al., 2013a*) across LC voltage waveforms (*Figure 2B,C*, time point II). Following this reduction, waveform synchrony values consistently recovered towards

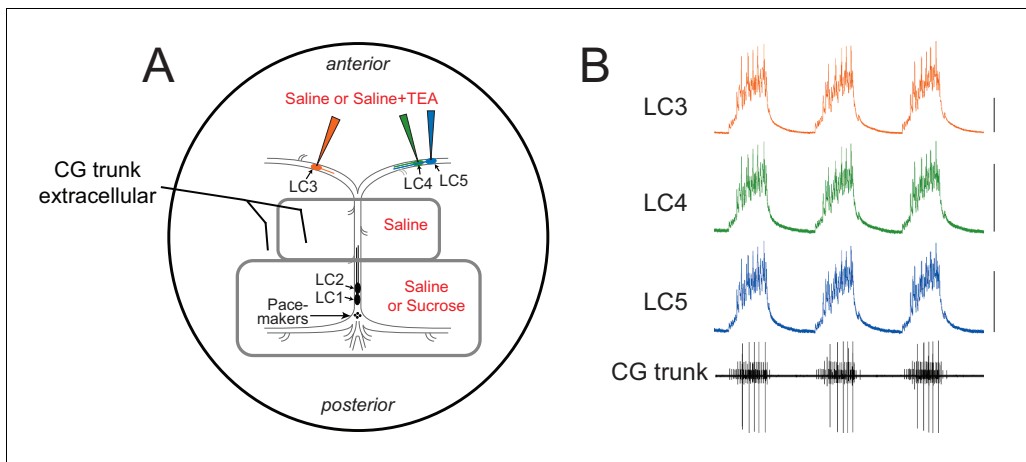

**Figure 1.** Experimental setup for the recording and superfusion of CG neurons. (**A**) Petroleum jelly wells (gray) allow the posterior LC1 and LC2 as well as the pacemaker small cells (SCs) to be pharmacologically isolated from the anterior large cells (LC3, LC4, LC5). Pacemaker cells can be maintained in physiological saline, or the network can be temporarily shut down by replacing saline with 750 mM sucrose. Extracellular recordings are performed with stainless steel pin electrodes from the 'trunk' nerve that contains the axons of all 5 LCs and the pacemaker cells. Intracellular recordings are taken from the anterior LCs. The area outside the petroleum jelly wells is superfused with pharmacological agents to target only the anterior large cells. (**B**) Simultaneous intracellular recordings from the three anterior LCs and extracellular recording of the network output via the trunk nerve, demonstrating synchrony among LCs in the control ongoing rhythm. Scale bars = 10 mV, recording duration = 9 s.

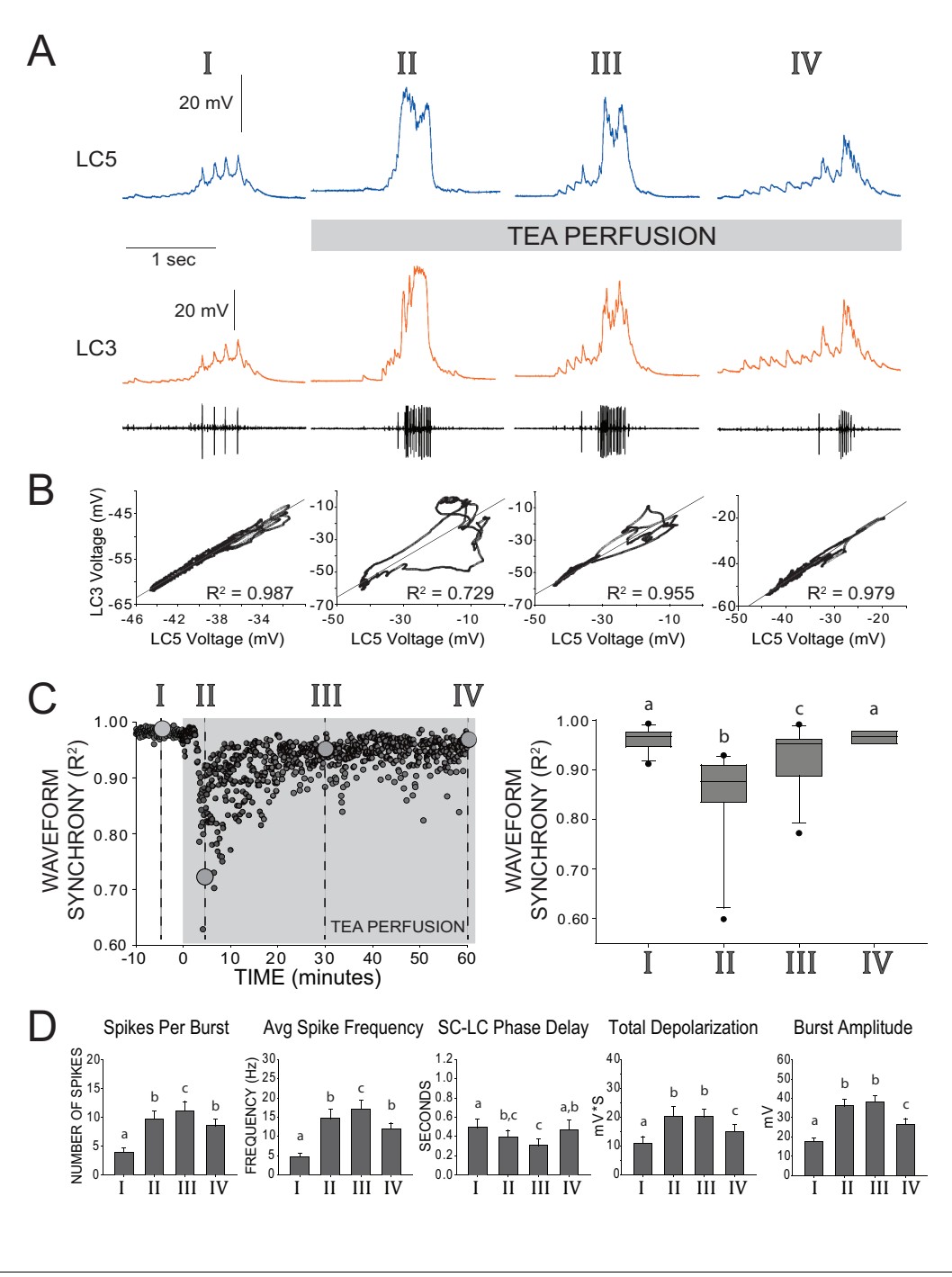

**Figure 2.** Restored excitability and waveform synchrony among LCs after 1 hr of TEA exposure. (**A**) Representative recordings of LC3 and LC5 and of network activity over one hour of TEA exposure. Roman numerals (I, II, III, IV) designate time points of reference throughout the remainder of the figure, as follows: I – control saline, II – acute TEA exposure identified as the maximum effect on loss of synchrony across LCs (i.e. lowest $R^2$ value), III – 30 min of TEA exposure, IV – 60 min of TEA exposure. (**B**) Scatterplots show pairwise correlation of time-matched voltages (sampled at 10 kHz) of the waveforms shown in the representative traces. $R^2$ values are calculated from Pearson's correlation tests for these two cells. Loss and restoration of conserved output is demonstrated by changes in coherence in the scatterplot as well as in $R^2$ value. (**C**) Synchrony of waveforms of the two cells seen in panels A and B plotted as $R^2$ values over the entire time course of the experiment. Roman numerals and large gray circles represent the values that were obtained from the scatterplots as each time point shown in panel B. TEA perfusion persists from time zero through 60 min. Box plots show distributions of $R^2$ values from cross-
*Figure 2 continued on next page*

*Figure 2 continued*

correlation analyses of LC voltage waveforms for pairs of LCs from N = 11 preparations. Lines within boxes mark the median, box boundaries represent 25th and 75th percentiles, whiskers represent 5th and 95th percentiles, and points represent outlying observations. Groups with significant differences in median synchrony (p<0.05; Wilcoxon signed rank tests) are denoted with different letters. (D) Excitability of LCs was quantified by five measurements (mean ± SD). Analysis of each preparation used the average of 10 consecutive bursts at each time point (N = 8 preparations). Number of spikes per burst, spike frequency within each burst, and the latency between pacemaker firing and first motor neuron spike (SC-LC Phase Delay) were calculated from extracellular traces. Total depolarization and amplitude of each burst are based on intracellular recordings. Significant differences across groups (p<0.05; paired *t*-tests) are denoted with different letters, such that any two bars with a letter in common are not significantly different.

baseline levels, often by 30 min, and were no longer statistically different from baseline by 1 hr of treatment with TEA (*Figure 2C*, right). To monitor compensatory changes in excitability, five different measures of excitability were calculated using both extracellular and intracellular recordings (*Figure 2D*; see Experimental Methods). The spikes per burst, average spike frequency, total burst depolarization and burst amplitude all were significantly increased immediately after exposure to TEA ('acute'), while the small cell pacemaker (SC)-to-LC phase delay was significantly decreased (*Figure 2D*). All 5 measures also then showed a significant change back towards their baseline levels between the 30 min and 60 min time points. While all measures showed clear shifts towards restoration of baseline excitability, the number of spikes per burst, spike frequency within each burst, burst amplitude, and total burst depolarization were not completely restored to control levels; the exception was the SC-LC phase delay which was fully restored (*Figure 2D*). Preparations exposed to TEA for 2–3 hr showed no further change in excitability (data not shown). For this reason, time scales longer than 1 hr were not included in our analyses.

## Modelling predicts compensation based on intrinsic conductances

TEA exposure reduces LC synchrony and induces hyperexcitability. Our model development and selection criteria resulted in a population of 27 model CG networks with variable underlying conductances of the constituent neurons that successfully recapitulated the biological data observed in TEA (see Methods, Supplemental Information). Our previous results identified an approximate 2.2 ± 0.8-fold change in $I_A$ in LCs as a result of 60 min of TEA exposure (*Ransdell et al., 2012*). Therefore, we used the model networks to explore potential mechanisms of compensation by first increasing and decreasing each individual maximal membrane conductance by a similar factor of 2. We searched for changes that would increase LC spike synchrony while countering the hyperexcitability induced by TEA. To easily visualize the trends, each network was normalized to its initial value for spike synchrony. These data are shown for all conductances in *Figure 3*.

Our initial goal with the model was to determine whether changes in single conductances were sufficient to elicit compensatory changes in output that help restore both excitability and synchrony. While it is not difficult to conceive of a change in multiple aspects of the parameter set that could achieve restoration of output, it is perhaps not as intuitive – but presumably the most parsimonious solution – for a single conductance to have such an impact. True to this expectation, while various manipulations of $G_{max}$ values improved either excitability or synchrony, very few conductance changes improved both. The optimal solution of significantly improving spike synchrony and also significantly decreasing the total number of action potentials was achieved in only one case: two-fold increase in $G_A$ resulted in a mean synchrony score that was significantly different from the TEA case (p<0.05, paired *t*-test) but not significantly different from control (p=0.157). No other change in a given conductance resulted in this combination of statistical outcomes. Not every model cell or networked improved uniformly with this conductance change. Therefore, while these results do not rule out a contribution for other conductances, they do suggest that an increase in $G_A$, as seen in previous experimental studies on isolated LCs (*Ransdell et al., 2012*), may be the most likely candidate for a change in intrinsic conductance promoting synchrony at the network level. These data suggest that while a single conductance change (increased $G_A$) can help restore both excitability and synchrony, variations in a single voltage-dependent conductance may not be sufficient to account for

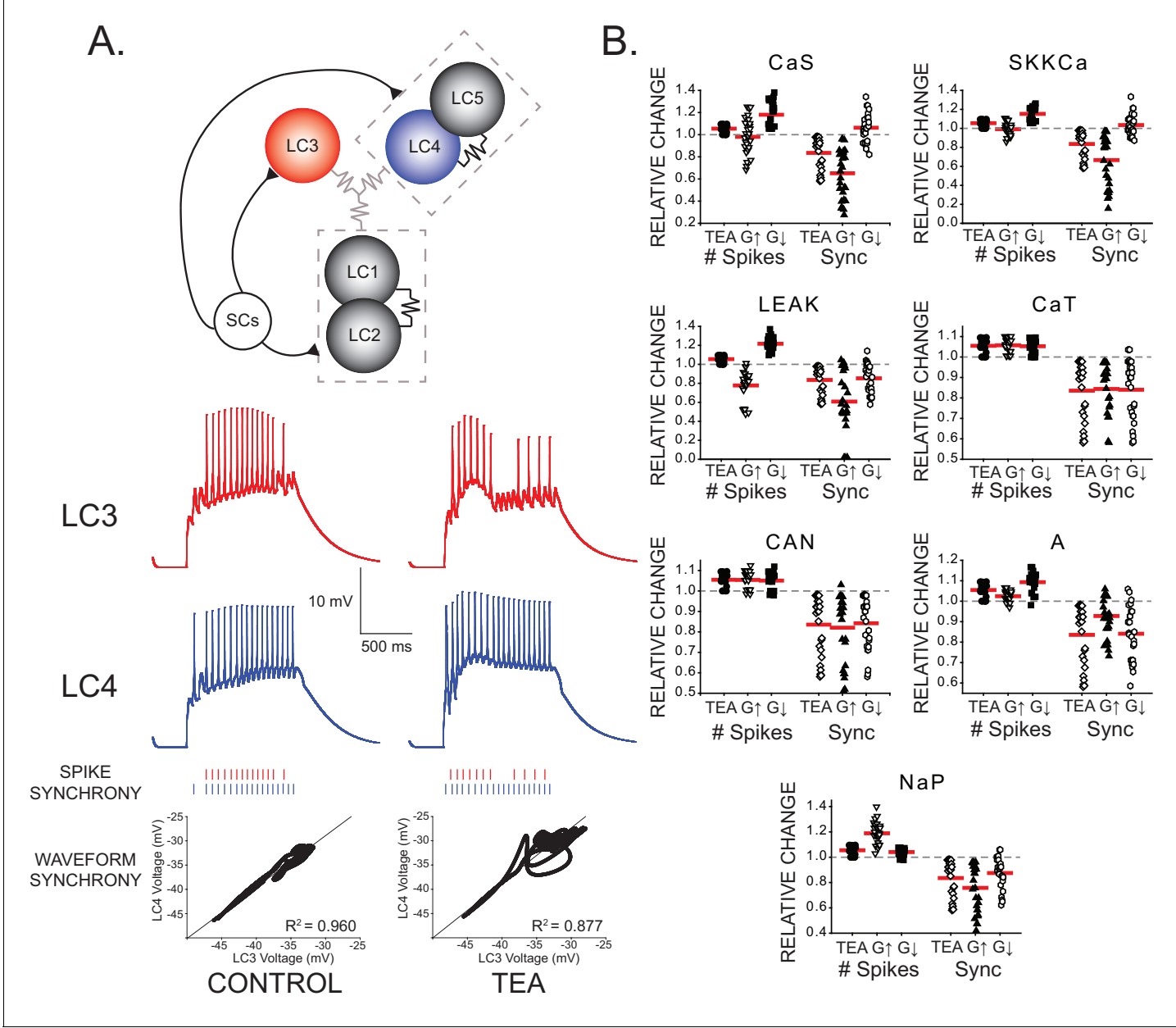

**Figure 3.** Effects of increasing and decreasing individual ionic conductances on excitability and synchrony in model CG networks. (**A**) Schematic representation of model network organization and connectivity. Five large cell (LC) motor neurons are innervated via excitatory synapses from a common small cell pacemaker input (SCs). LC model neurons consist of two compartments - soma and axon - of which only the somata are pictured. Somata contain 9 conductances: $G_{CaS}$, $G_{CaT}$, $G_{LEAK}$, $G_{CAN}$, $G_A$, $G_{BKKCa}$, $G_{SKKCa}$, $G_{Kd}$, and $G_{NaP}$. Paired LCs (1+2, 4+5) have stronger local coupling (black resistor symbols), and all 5 LCs are reciprocally electrically coupled via weaker gap junctions (gray resistor symbols). An example of LC3 and LC4 model output within a network burst activity is shown in the red and blue traces under both control and TEA (90% reduction in both $G_{Kd}$ and $G_{BKKCa}$) conditions. Graphical representations of spike synchrony (raster plots) and waveform synchrony (scatterplots; as in *Figure 2*) are shown for the model neurons, demonstrating that both measures reflect the loss of LC synchrony as a result of TEA. (**B**) Measurements of both output variables (# of spikes and spike synchrony) were made under three model conditions: control, TEA, and TEA + either a 2x increase (G↑) or 2x decrease (G↓) in a given conductance. N = 27 distinct model networks. All output measurements are normalized to their initial (control) conditions. Red lines represent the mean for a given group. Dashed line represents the 1.0 value (baseline) for a given measure. Compensatory responses that restore excitability and synchrony will tend to move the mean towards baseline.

the full compensation response. In addition to perturbing only individual conductances, we also varied current kinetics and activation parameters (half-activation voltage $V_{1/2}$, ± 10 mV, and slope factor $k$, by 0.5x and 2x (*Ballo et al., 2010*) and time constant by ± 10 ms) for all the cell currents individually, and found that no changes in parameters for a single current could simultaneously restore excitability at the single cell level, and synchrony at the network level (data not shown). While the analysis has focused only on the parameters of a single current, simultaneous changes in parameters of multiple currents could also potentially provide similar compensation, and that remains to be explored. However, our analysis does reveal the substantial contribution of changing a single parameter – $G_A$ – on multiple aspects of network compensation, to an extent that is beyond simple intuition. Importantly, the model also extends the biological data by demonstrating that waveform synchrony can translate into spike synchrony. Because of the electrotonic distance between the somata and axons of LCs, we cannot measure spike synchrony directly in this preparation. The model allows us to infer that waveform synchrony (and loss of synchrony) can indeed translate to the level of the most proximal cellular output – spiking.

## Intrinsic compensation contributes to restoration of synchrony

Model runs predicted that increases in $I_A$ help restore LC excitability and synchrony. To test this experimentally, we silenced pacemaker activity with isotonic sucrose solution (see Methods, Supplemental; *Figure 1A*), and tested the similarity of responses of each individual LC to a biologically realistic current stimulus (*Ransdell et al., 2013a*). We compared LC3 and LC5 to the same current injection at three time points: control, 5 min post-TEA, and 1 hr post-TEA. Between current injections, pacemaker activity was restarted by removal of the sucrose block. This allows us to test each cell in isolation, but compensation occurs in the intact network. The initial voltage responses to our stimulation protocol in LC3 and LC5 in control conditions are highly similar to one another, and their level of waveform synchrony was not significantly different from the synchronous activity across these LCs during intact control network activity (*Figure 4A,B*). Immediately following TEA application, LC3 and LC5 show disparate output when driven with a common stimulus protocol (*Figure 4A*). Finally, our data show significant increases in $R^2$ of voltage activity within 1 hr across isolated LCs (*Figure 4B*), demonstrating that *intrinsic* compensation does improve *network* synchrony. However, after 1 hr the synchrony values were significantly lower than control values (*Figure 4B,C*), suggesting that intrinsic compensation alone is insufficient to restore synchrony. To determine whether compensatory changes in $I_A$ occur in the intact network, we measured $I_A$ with two-electrode voltage clamp in LCs before and after 1 hr of TEA exposure. Measurements were made while the network activity was temporarily halted with sucrose block, and compensation occurred with ongoing network activity. In all cases peak $I_A$ current increased (*Figure 4D*), with a mean increase of 56% ($p<0.05$, $n = 6$, Wilcoxon signed rank test). These data are consistent with the hypothesis that a compensatory increase in $I_A$ can help promote synchrony in these networks.

Although the waveforms of LC3 and LC5 were not different from one another after 1 hr of compensation, anterior LC burst potentials did not reproduce their *original* waveform after compensation (see *Figure 2*; Panels I and IV). We also used the LCs from data shown in *Figure 4A* to compare the voltage responses of individual LCs to a fixed stimulus before and after compensation. Repeatable voltage responses under control conditions indicate that trial-to-trial variability is negligible (mean $R^2 = 0.997$; *Figure 4E*, control). However, the voltage response after compensation was significantly different from the control voltage response ($p<0.01$, $n = 8$, Wilcoxon signed rank test), indicating that intrinsic compensation does not restore the original cellular output (*Figure 4E*; control vs. TEA [1 Hour]).

## Increased model electrical coupling conductance helps restore synchrony

If intrinsic compensation does not fully restore synchrony, another mechanism must be present to explain the results observed during network compensation. LCs receive common excitatory inputs from the pacemakers and one hypothesis is that changing the strength of these chemical synapses might help to restore LC firing to appropriate levels. LCs in the network are also electrically coupled to one another via gap junctions which presumably promotes synchrony, although clearly the native coupling is not able to maintain LC synchrony in TEA (*Hagiwara et al., 1959*; *Tazaki and Cooke,*

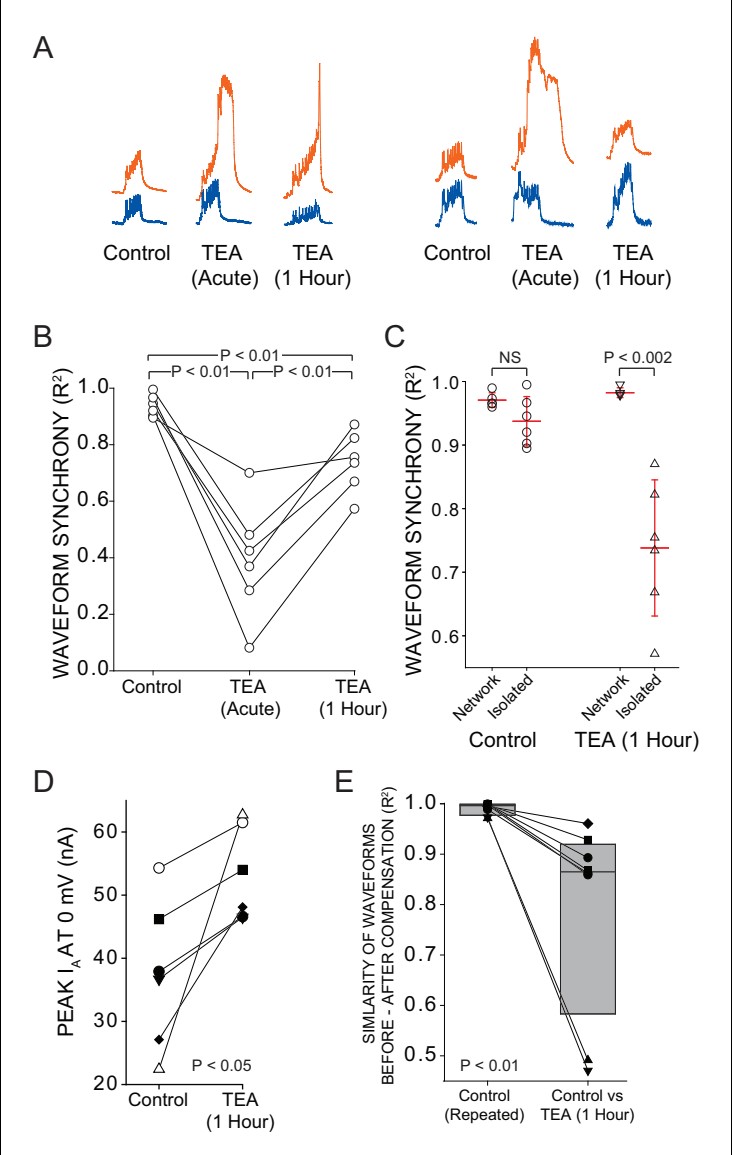

**Figure 4.** Intrinsic compensation involving $G_A$ partially restores synchrony after TEA block. A reversible sucrose block was used to temporarily stop network activity and a current stimulus protocol was delivered to individual LCs at three time points: control, after 5 min of TEA perfusion (acute), and after 1 hr of TEA perfusion (after compensation). (**A**) Representative traces from 2 different preparations compare the responses of LC3 (orange) and LC5 (blue) to the same current injection (Stimulus Protocol) at each of the three time points. (**B**) $R^2$ values for N = 6 preparations at each time point are plotted with the same preparation connected across time points. All 3 conditions were significantly different from one another (p<0.01; paired *t*-tests). (**C**) When isolated cell output was compared with the output in the network, there was no difference in $R^2$ at the control state, but following 1 hr of compensation in the network, there was a significant (p<0.002 – *t*-test) difference in synchrony scores (mean ± SD) between cells when isolated vs. when they are in the intact network. (**D**) $I_A$ was measured by two-electrode voltage clamp before and after compensation in N = 6 LCs in the intact CG. Voltage clamp data were obtained by temporarily silencing network activity with 750 mM sucrose. There was a significant increase in $I_A$ after 1 hr in TEA (mean 56 ± 65% increase, p<0.05 –Wilcoxon signed rank test). (**E**) Similarity of waveform in the same neurons before and after 1 hr TEA exposure. $R^2$ values were calculated for the output of the same cells before and after TEA exposure, and are shown as before-and-after values in the same cell connected by a line. Box plots show distributions of $R^2$ values from cross-correlation analysis of LC voltage waveforms N = 8 cells. Lines within boxes mark the median, box boundaries represent 25th and 75th percentiles. 'Control' is the comparison of voltage waveforms to 2 separate rounds of current injection in the absence of TEA. Although there was improvement in similarity of waveforms after 1 hr in TEA (panel A), the newly compensated output after 1 hr of TEA exposure does

*Figure 4 continued on next page*

*Figure 4 continued*

not recapitulate the original response to the stimulus protocol (significantly different from control; p<0.01 – Wilcoxon signed rank test).

*1983*; *Cooke, 2002*). A second hypothesis is that increased electrotonic coupling between LCs could buffer against disparate output and help to restore synchrony.

Using our set of model networks, we increased and decreased the strength of chemical synapses in 10% increments to test the effects on excitability and synchrony. We then did the same with model electrical coupling conductance. We found that increasing the strength of either chemical synapses or electrical coupling increased both synchrony and excitability (*Figure 5 left*). However, increasing the chemical synaptic conductance in conjunction with TEA blockade also increases spiking of the LCs ~25–30% in contrast to the biological decrease in excitability relative to the acute TEA exposure seen with compensation. Conversely, only a small change in LC spiking occurs with an increase in electrical coupling (~9%, *Figure 5 left*). Reducing the strength of either chemical synapses or electrical coupling decreased overall spike synchrony (*Figure 5 right*), violating the assumptions of compensation based on the biological data. Reducing chemical synaptic strength eventually ceased LC firing altogether (data not shown).

## Increased electrical coupling helps restore synchrony across LCs in experiments

Increased electrical coupling restored synchrony in model LCs with only a modest effect on excitability. To investigate this relationship in experiments, we isolated LCs and used a dynamic clamp to

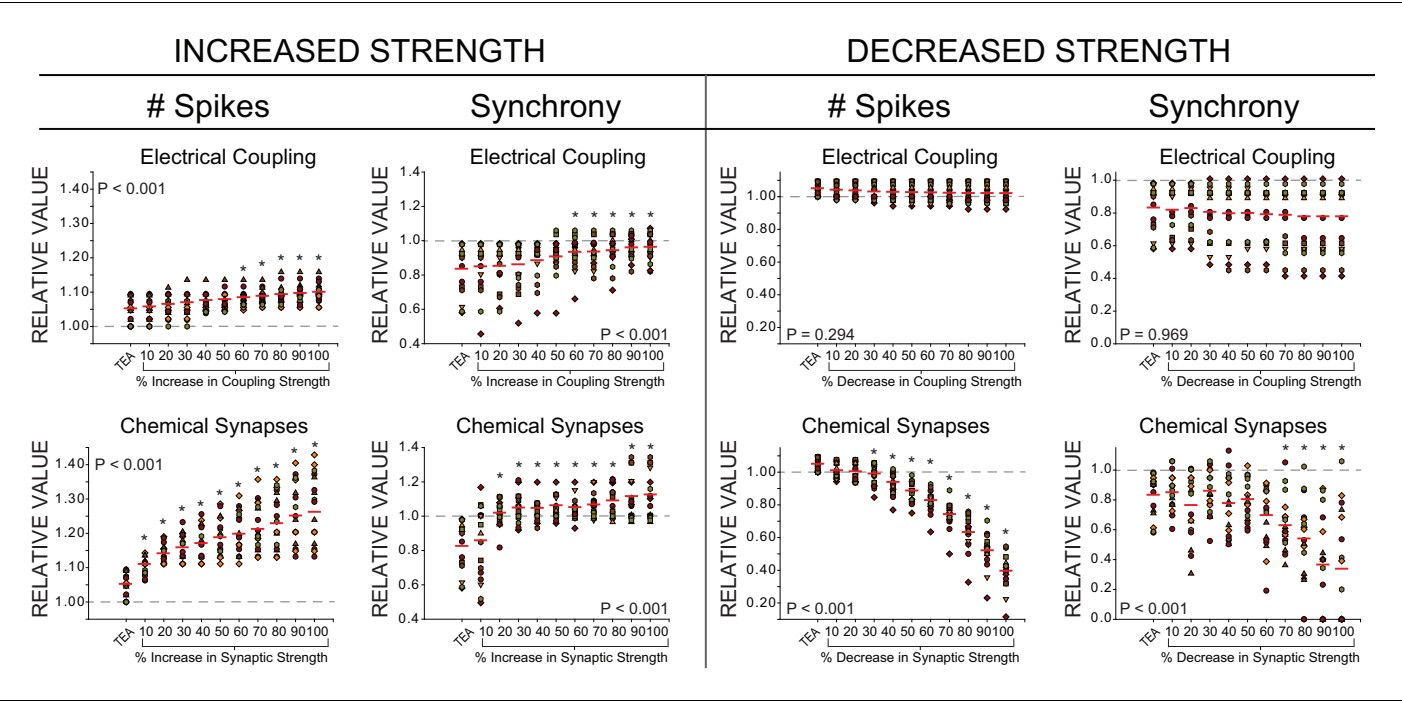

**Figure 5.** Effects of increased or decreased strength of chemical synapses and electrical coupling on excitability and synchrony in model CG networks. Measurements of two output variables (# of spikes and synchrony) were made under three model conditions: control, TEA (90% reduction in both $G_{Kd}$ and $G_{BKKCa}$), and TEA + an incremental increase or decrease (up to 100% by 10% increments) for both chemical (pacemaker to LC) or electrical (LC to LC) connections. N = 27 distinct model networks. All output measurements are normalized to their initial (control) conditions to visualize trends. Dashed line represents the 1.0 value (baseline) for a given measure. Red lines represent the mean for each group. Each different colour and shape for points corresponds to one model network, and the same networks are shown across conductance levels. P-values in each plot refer to the results of a one-way ANOVA across all groups. Asterisks (*) denote groups in each plot that were significantly different from the TEA group via Holm-Sidak post-hoc tests.

add an artificial coupling conductance. Pairs of LCs from the same network were physically isolated by thread ligature, exposed to TEA, and simultaneously received the same stimulus protocol, while the dynamic clamp added a non-rectifying artificial coupling conductance (from 0 to 0.2 µS) between the cells. The driving force was equivalent to the voltage difference in membrane potential between the coupled cells. Increasing the artificial coupling conductance significantly increased the correlation coefficient of the waveform between the two cells (*Figure 6A*), with a synaptic conductance value of 0.2 µS able to rescue synchrony of LCs to levels observed in intact networks (*Figure 6A*). These results provided proof of principle that increasing electrical coupling could be responsible for resynchronization in the network.

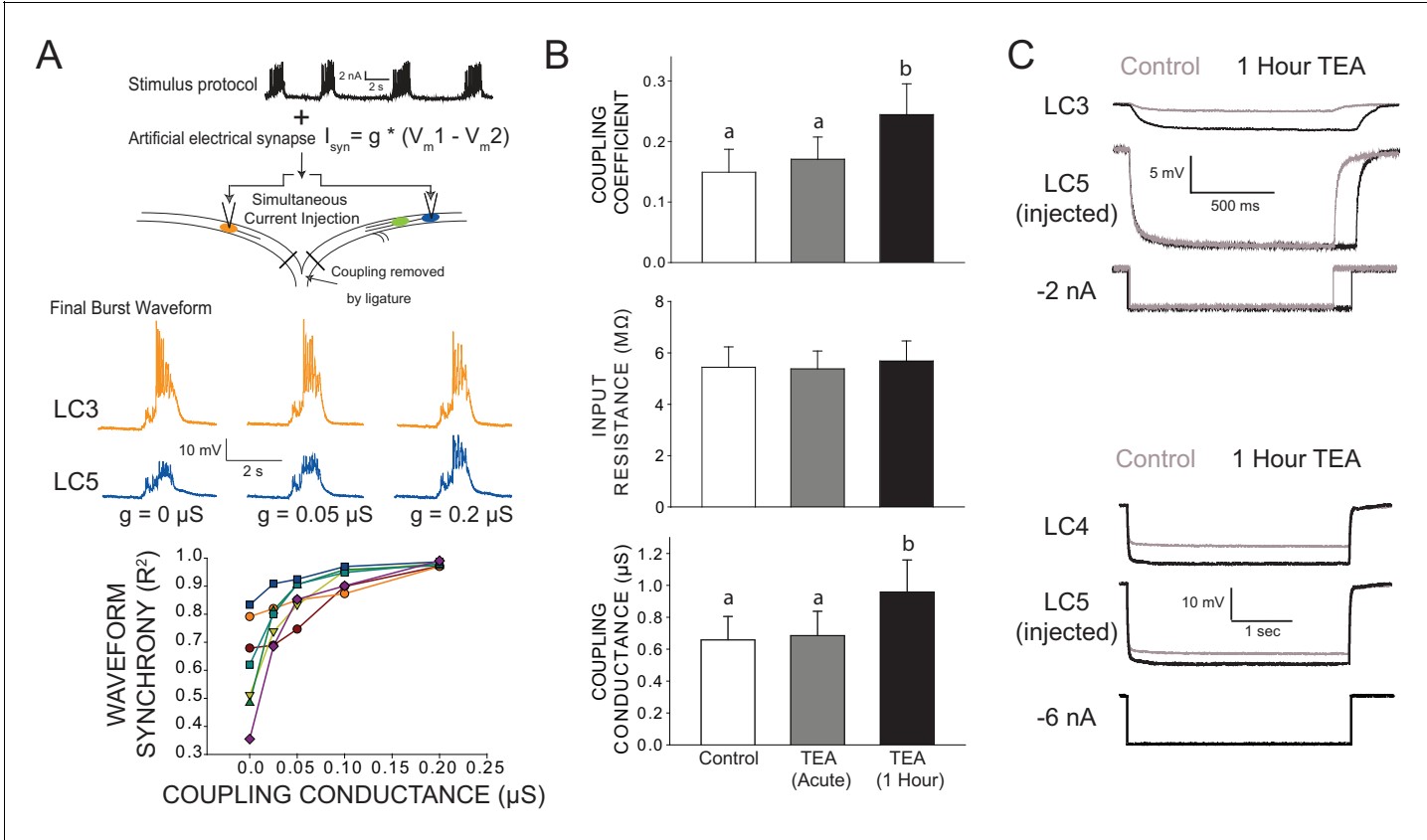

**Figure 6.** Changes in electrical coupling associated with compensation in biological CG networks. (A) Artificial electrical coupling restores synchrony in isolated LCs. With acute exposure to TEA, isolated LCs produce disparate output in response to an identical stimulus. The stimulus protocol consists of current injections that mimic biological synaptic currents and back propagating action potentials (see *Ransdell et al., 2013a*) from four consecutive bursts of network activity. The current was injected simultaneously into isolated cells, while the dynamic clamp was used to provide an artificial coupling conductance. Representative traces of the same two cells shown in TEA with different levels of synaptic current applied. Only the final burst of the four-burst input stimulus is shown for clarity. N = 7 different preparations (bottom panel) show an increase in synchrony with increasing coupling conductance. (B) Biological coupling increases during network compensation. Hyperpolarizing current injections were used to measure coupling coefficients between LC3 and LC5 during control conditions, with acute TEA exposure (5 min) and after 1 hr compensation in TEA. The input resistance of LCs was measured and showed no significant change at any time point. No significant differences were observed for coupling coefficient or coupling conductance between control and acute TEA conditions. Coupling coefficient significantly increased after 1 hr in TEA (Mean increase 85 ± 82% from control, N = 11, p<0.01, Wilcoxon signed rank test). Measurements from LC4 and LC5 show that coupling conductance increased significantly as a result of 1 hr TEA exposure (mean increase 49.5 ± 36% from control, N = 13, p<0.001, Wilcoxon signed rank test). Significant differences across groups are denoted by different letters. Plots show mean ± SD. (C) Representative traces from two different preparations of changes in coupling observed before (Control) and after 1 hr of TEA exposure. Top traces are between LC3-LC5 and the bottom traces are between LC4-LC5. Measurements were made in two-electrode current clamp, and the current was injected into LC5. Because current injections were manually timed to occur between bursts of network activity, slightly different durations of current pulses occurred in the two recordings in the top recordings. Recordings from LC4-LC5 were used for coupling conductance measurements seen in panel B, as their close proximity allows for much less influence of electrotonic distance on calculations of conductance.

We then measured coupling coefficients between LCs during compensation in the intact network. Coupling coefficients between cells increased significantly after 1 hr in TEA (mean increase 85%, p<0.01; *Figure 6B,C*). The coupling coefficient is a useful description of the functional relationship in coupling, but does not identify the electrophysiological mechanism. Plasticity of coupling properties can ultimately be influenced by two fundamentally different mechanisms: altered resistance of the non-junctional membrane of the coupled cells, or modification of gap junctional conductance. Using two-electrode current clamp, we saw no significant differences in the apparent input resistance of LCs in control physiological saline, after acute TEA exposure, or after compensation (1 hr TEA exposure; *Figure 6B,C*). These results indicate that changes in passive membrane conductance ($G_{Leak}$) are not responsible for increased coupling coefficients.

To directly test whether coupling conductance between LCs increases as a result of TEA-induced compensation, we focused on LC4 and LC5 pairs within the same network. In the crab CG, anterior LCs exhibit strong local electrical and dye coupling (*Tazaki and Cooke, 1979*, *1983*). The branch containing LC4 and LC5 somata can be separated and electrotonically 'sealed' from the network by thread ligature to create ideal conditions for measuring coupling conductance. With two electrodes in each cell, we used hyperpolarizing current injections to measure resistance and calculate junctional conductance independent of membrane resistance (as in *Bennett, 1966*; see Materials and methods). Coupling conductance ($G_C$) between LC4 and LC5 significantly increased during 1 hr of TEA exposure (mean increase 49.5%, p<0.01, N = 8; *Figure 6B,C*).

## Interaction of intrinsic and electrical synaptic compensation

Taken together, our experimental and modelling results suggest that an increase in $G_A$ is able to counter the increase in excitability of LCs in TEA in a compensatory fashion, as well as promote restoration of synchrony, but was insufficient to restore synchrony fully. Additionally, our results suggest that an increase in coupling among LCs can greatly promote synchrony with only a modest effect on excitability. Therefore, we next used our model networks to investigate how $G_A$ and $G_C$ might interact to promote synchrony by calculating synchrony scores as conductances of all 27 model networks were adjusted. First, we increased $G_A$ alone in 10% increments up to a 100% increase (*Figure 7*). Increasing $G_A$ up to +40% promoted greater synchrony after TEA blockade, but was unable to fully restore synchrony even with increasing conductance levels, consistent with our biological data (*Figure 4*). Increasing $G_A$ beyond +40% did not further improve synchrony (*Figure 7*), and ultimately caused LCs to cease firing altogether. We also increased model $G_C$ incrementally (from +10% to +150%), and found that electrical coupling alone was capable of restoring synchrony fully, but this required a 140% increase in its value (*Figure 7*). Finally, we increased both $G_A$ and $G_C$ together in 10% increments, revealing a potentially synergistic relationship: a smaller increase of 70% in each conductance was able to produce spike synchrony that was indistinguishable from control (*Figure 7*).

## Discussion

Homeostatic or compensatory plasticity in the nervous system has been the subject of intense interrogation, with studies focusing on both homeostatic synaptic scaling (*Turrigiano, 2012*; *Lee et al., 2014*) and tuning of ionic conductance relationships to maintain a target level of excitability (*Turrigiano et al., 1994*; *Desai et al., 1999*). However, few studies have sought to integrate multiple mechanisms to directly address emergent network stability from compensatory processes acting at the level of single neurons. Moreover, only recently has there been an appreciation that variability in underlying cellular parameters such as conductance magnitudes (*Schulz et al., 2006*; *Pratt and Aizenman, 2007*; *Wilhelm et al., 2009*), activation properties of channels (*Olypher et al., 2006*; *Amendola et al., 2012*), and synaptic strengths (*Olypher and Calabrese, 2007*; *Wilhelm et al., 2009*; *Grashow et al., 2010*) may form part of the repertoire of compensatory mechanisms that endow networks with remarkable robustness characteristics. Homeostatic plasticity has been reported in single isolated neurons (*Swensen and Bean, 2005*; *Ransdell et al., 2012*), in artificial networks formed in culture (*Desai et al., 1999*; *Ibata et al., 2008*), or in thin sections of the CNS (*Karmarkar and Buonomano, 2006*; *Lambo and Turrigiano, 2013*).

Cortical slice preparations have demonstrated homeostatic plasticity of intrinsic conductances coordinated with chemical synaptic plasticity (*Karmarkar and Buonomano, 2006*; *Lambo and*

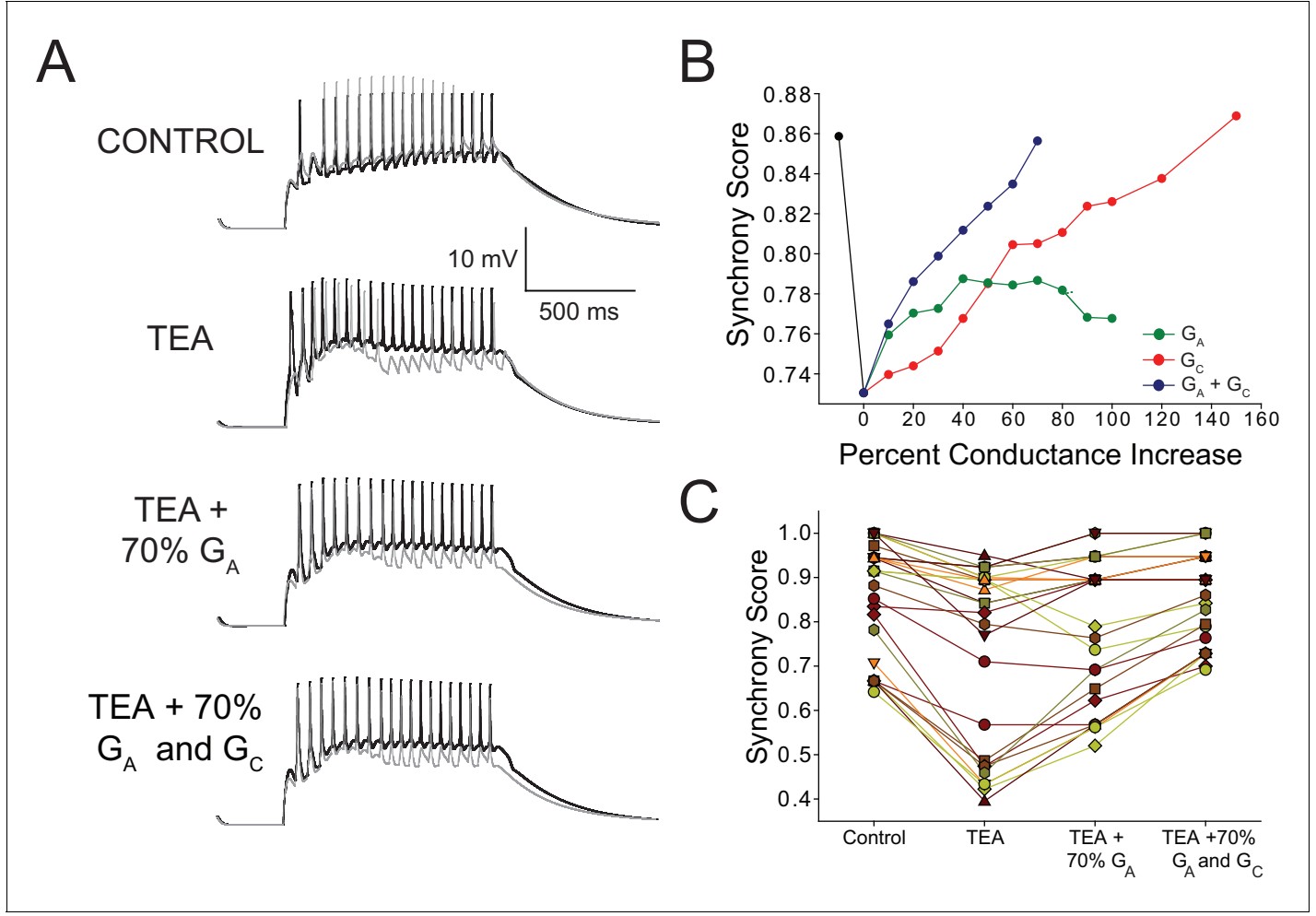

**Figure 7.** Increased $G_A$ and coupling conductance ($G_C$) among LCs act in concert to help restore synchrony across LCs in model networks. (**A**) Voltage response of a typical network LC3 (gray) and LC4 (black) cells under three model conditions: control, TEA (90% reduction in both $G_{Kd}$ and $G_{BKKCa}$), TEA + 70% increase in $G_A$, and TEA + 70% increase in $G_A$ and $G_C$. (**B**) Effects of increasing $G_A$ alone, $G_C$ alone, or both $G_A$ and $G_C$ on synchrony in model networks. Dashed lines represent simple linear regression fits to the points for each condition. Black line represents the change in synchrony from control to TEA. Points shown are the average values for N = 27 networks. (**C**) Summary of Synchrony Score shown for all 27 model networks. Stepwise changes are shown from Control, TEA, TEA + increasing $G_A$ by 70%, and TEA + increasing both $G_A$ and $G_C$ by 70% (the point at which maximal synchrony is restored as per the analysis in panel B). Individual points correspond to those used to generate averages in panel A at the 70% level to give an idea of the variability in the data set. Individual preparations are connected with lines to show trends across networks.

*Turrigiano, 2013*), and Mauthner cells in fish have been shown to exhibit coordinated activity-dependent changes in both the chemical and electrical components of its mixed synapses (*Yang et al., 1990*; *Pereda and Faber, 1996*). While membrane conductances and properties of electrical coupling are known to interact in critically important ways to promote synchrony (*Curti et al., 2012*; *Gutierrez and Marder, 2013*), to our knowledge the present study is the first demonstration of coordinated homeostatic plasticity of intrinsic and electrical synaptic conductances in a comprehensive network-level example of homeostatic compensation.

We hypothesized that plasticity distributed throughout the network might provide robustness of network output. Using a combination of pharmacology, electrophysiology, and modelling approaches in intact neural networks, we first uncovered a striking vulnerability of neural networks to underlying cellular variability. We then demonstrated that an intact network has mechanisms that allow for robustness in the output via synergistic compensatory changes across the individual cell (ionic conductance) as well as trans-cellular (electrical coupling) properties. The interplay of intrinsic and synaptic parameters, and distributed plasticity in determining network output, is difficult, if not

impossible, to unravel experimentally. This makes investigations via biophysical models an attractive alternative to narrow possibilities, and complement experiments, as we have demonstrated.

## Multi-component compensation can synergistically restore network output

Previous modelling studies found that K$^+$ currents can increase or help restore synchrony between electrically coupled neurons (*Pfeuty et al., 2003*), so we first hypothesized that a compensatory increase in A-Type K$^+$ membrane conductance could be a mechanism underlying both restored excitability and resynchronization. Over the course of 30–60 min, increased I$_A$ was associated with decreasing cellular excitability [see also (*Golowasch et al., 1999*)] and improvement of coordinated motor neuron firing. However, intrinsic compensation alone was insufficient to fully restore synchrony across LCs. A concomitant increase in electrotonic coupling ensured virtually complete resynchronization. Our modelling results suggest that although a sufficient increase in electrical coupling alone could restore full synchrony (140% increase), it could not simultaneously restore the original level of excitability. Only a 70% increase was necessary when accompanied by a concomitant increase in G$_A$. Therefore, we conclude that multi-component mechanisms are not only necessary for full compensation, but also that their synergistic action is potentially more efficient than either mechanism operating in isolation.

## Comparison of our results in the context of behavioural/motor plasticity

While our results occurred in a compensatory context, the underlying mechanisms bear striking similarity to motor output plasticity induced by operant conditioning in *Aplysia*. In a series of elegant experiments, it has been shown that chaotic exploratory and consummatory radula biting movements of *Aplysia* during food searching behaviour can be stably modified by operant conditioning, leading to prolonged bouts of radula movements with increased frequency and more stereotyped rhythmic organization (*Nargeot et al., 2007*; *Nargeot and Simmers, 2011*). Chaotic biting patterns result from inherently variable and unsynchronized bursting of CPG neurons that are each randomly capable of triggering bites (*Nargeot et al., 2009*). Following operant conditioning, induction of regular and synchronized bursting of pattern-initiating cells can be attributed to changes in both intrinsic excitability and electrical coupling strength. Specifically, changes in intrinsic excitability attributed to changes in leak conductance underlie the increase in frequency of motor output, while increases in coupling strength allow for the synchronization and regularization of bursting (*Nargeot et al., 2009*; *Sieling et al., 2014*). The full shift in behavioural and circuit output is therefore the additive influence of both intrinsic and electrical synaptic conductances. The striking similarity in these underlying mechanisms suggests that these kinds of circuit-level mechanisms may be a conserved strategy for stabilization of synchrony within network output, be it compensatory or in the context of behavioural plasticity.

## Physiological regulation of coupling conductance

The speed (within 30 min) and magnitude (up to a doubling of effective coupling) of physiological changes seen in electrical coupling was remarkable. Although electrical coupling has long been known to promote synchrony in many systems, including the CG (*Tazaki, 1972*; *Bennett and Zukin, 2004*), the physiological interaction of electrical coupling with intrinsic conductances to affect a compensatory output has not been examined. Previous work in the crustacean STG has demonstrated how the synchronized activity of pacemaker cells is dependent on an interaction of intrinsic conductances and electrical coupling (*Szücs et al., 2000*, *2001*; *Soto-Treviño et al., 2005*), and that distinct circuits can be brought into synchrony via manipulations of electrical and chemical synapses (*Elson et al., 1998*; *Szücs et al., 2000*, *2009*). But none of these studies have addressed the interaction of membrane conductance and electrical coupling in a compensatory context. Similarly, plasticity of electrical synapses has drawn considerable attention after being discovered in the mammalian central nervous system, including the thalamic reticular nucleus (*Landisman and Connors, 2005*; *Haas et al., 2011*), inferior olive (*Lefler et al., 2014*; *Mathy et al., 2014*), and retina (*Kothmann et al., 2009*; *Völgyi et al., 2013*). Studies in the thalamic reticular nucleus have suggested that potentially compensatory changes in coupling are important to maintain network

stability as large changes in intrinsic excitability occur across development (*Parker et al., 2009*). These discoveries increased awareness of the complex functional roles and plasticity of coupling (*Pereda et al., 2013*; *O'Brien, 2014*; *Haas, 2015*), and also spurred research to identify molecular mechanisms that underlie plasticity and maintenance of these structures (*Flores et al., 2012*; *Li et al., 2012*; *Turecek et al., 2014*). Our study adds to this growing appreciation for plasticity of electrical synaptic connections in the context of homeostatic plasticity.

### Variability, plasticity, and network output – the bigger picture

Stable levels of intrinsic neuronal excitability and temporal coordination within networks are critical features across all nervous systems. Underlying both neuronal and network outputs are complex, and often highly variable intrinsic and synaptic properties of constituent neurons (*Marder, 2011*; *Norris et al., 2011*). Our data demonstrate that the intrinsic variability between cells of the same type can make networks vulnerable to loss of temporal coordination, in this case desynchronization of motor neuron output. Although LC activity was fully resynchronized within 1 hr, recovered LCs never recapitulated their original voltage waveforms. While the intrinsic conductances involved in our manipulation and compensation ($G_{Kd}$, $G_{BKKCa}$, $G_A$) have overlapping functions and characteristics (*Ransdell et al., 2012*), our findings demonstrate that individual conductances are not truly redundant. Degeneracy of ion channel properties leading to this type of relationship has been put forth as a mechanism underlying robustness and adaptability in neural networks (*Tononi et al., 1999*; *Marder and Goaillard, 2006*), but our study suggests physiological limits to neural network compensation and robustness. These limits may themselves be a contributing factor to the nature and progression of pathology in neurodegenerative diseases (*Trasande and Ramirez, 2007*; *Beck and Yaari, 2008*; *Small, 2008*).

We induce desynchronization and compensation in our studies through pharmacological block of a subset of $K^+$ conductances with TEA. However, the precise role these mechanisms play in fully intact biological networks is unclear. Intrinsic conductances can be differentially affected by ubiquitous natural mechanisms such as neuromodulation (*Marder, 2011*, *2012*) or temperature changes (*Tang et al., 2012*; *Marder et al., 2015*). Further, the effectiveness of electrical coupling can be affected by the modulation of intrinsic cellular conductances (*Szabo et al., 2010*). Maintaining reliable synchronization of the output under changing conditions is not trivial, and understanding the robustness and the constraints of homeostatic systems that cope with such perturbations remains an important area for future investigation (*Marder et al., 2014*).

## Materials and methods

### Animals

Adult male Jonah crabs, *Cancer borealis*, were shipped overnight from The Fresh Lobster Company (Gloucester, MA). Crabs were maintained in artificial seawater at 12°C until used. Crabs were anaesthetized by keeping them on ice for 30 min prior to dissection. The complete CG was dissected from the animal and pinned out in a Sylgard-lined petri dish in chilled physiological saline (440 mM NaCl, 26 mM $MgCl_2$, 13 mM $CaCl_2$, 11 mM KCl, and 10 mM HEPES, pH 7.4–7.5, 12°C). Chemicals were obtained from Fisher Scientific unless otherwise noted.

### Biological methods: Electrophysiology

The CG network is comprised of 9 cells: 4 Small Cell (SC) pacemaker interneurons which give simultaneous excitatory input to 5 Large Cell (LC) motor neurons. Superfusate of SCs can be separated from the anterior LC somata using petroleum jelly wells (*Figure 1A*). Intact network activity was monitored with intracellular recordings from anterior LCs along with extracellular recording of the network output. For most experiments, the posterior end of the ganglion was maintained in normal physiological saline and protected from TEA superfusate with a barrier of petroleum jelly. The anterior end of the preparation was superfused at a rate of approximately 2 ml/min. A schematic of this experimental setup is shown in *Figure 1A*. All experiments were performed at 12°C.

Extracellular recordings using a Model 1700 Differential AC Amplifier (A-M Systems, Carlsborg, WA) were taken with stainless steel pin electrodes from a petroleum jelly well on the ganglionic trunk containing axons of all 9 neurons in the CG. LC spikes on the extracellular traces are easily

distinguishable by their large amplitude. The LC somata were desheathed for sharp electrode recordings. Intracellular recordings were made using glass electrodes containing 3 M KCl (8–25 MΩ) and AxoClamp 900A and AxoClamp 2B amplifiers (Molecular Devices, Sunnyvale, CA). Two-electrode voltage clamp (TEVC) and two-electrode current clamp (TECC) protocols were created and run using Clampex 10.3 software (Molecular Devices).

Somata were isolated for dynamic clamp experiments by tightening a thread ligature past the anterior branch point on the nerve containing the LC soma. Isolated cells were simultaneously driven with the same current stimulus, described previously (*Ransdell et al., 2013a*). Briefly, a stimulus protocol was generated by recording the voltage waveform from a LC somata during intact network activity. This consisted of a 20 s recording from a LC3 soma which included four burst potentials with both pacemaker cell EPSPs and LC back-propagating APs present. In addition to the stimulus current, dynamic clamp artificial coupling current was applied with NetClamp software (developed in the Fishberg Department of Neuroscience of the Mount Sinai School of Medicine and available at http://gothamsci.com/NetClamp/) at a sampling rate of 50 kHz according to the equation: $I_{gap} = g^*$ $(V_m1 - V_m2)$ where g is a non-rectifying coupling conductance under experimental control, and the voltage difference between the cells ($V_m1 - V_m2$) determines the driving force.

$I_A$ was measured before and after compensation using voltage clamp protocols as described previously (*Ransdell et al., 2012*). Briefly, outward currents were measured from a holding potential of −30 mV and stepped from −50 mV to +5 mV in 5 mV increments in order to measure the high threshold $K^+$ current $I_{HTK}$ which is blocked by TEA. In LCs, $I_{HTK}$ is predominantly a mix of BKKCa and delayed rectifier currents (*Ransdell et al., 2012*). A-Type $K^+$ current ($I_A$) was measured by performing an identical voltage clamp steps from a holding potential of -80mV and subtracting $I_{HTK}$. P/N leak subtraction was used for all TEVC. Coupling in the intact network was measured using TECC in both cells during the sucrose block. Negative current steps ranging from 1–6 nA were injected into one cell at a time while measuring voltage changes in both cells. Coupling coefficients were calculated as the ratio: ($\Delta V_{coupled\ cell}$ / $\Delta V_{Injected\ Cell}$).

## Biological methods: Measurements of intrinsic compensation

To determine whether intrinsic compensation may be contributing to restoration of synchrony in the biological network, we used a current stimulus protocol simulating realistic network inputs to LCs in order to deliver the same biologically relevant stimulus at 3 time points (*Ransdell et al., 2013a*). Using this reversible sucrose block to suspend pacemaking activity, we were able to compare the similarity of responses of LC3 and LC5 to the same current injection at three time points. This allows us to test each cell in isolation with respect to its output waveform, but compensation occurs with full network activity after removal of the sucrose over the course of 1 hr. We measured individual LC responses to current injection in control saline, repeated after 5 min of TEA exposure (acute), and again after 1 hr of TEA exposure.

Our modelling results suggested that an increase in $I_A$ may act in a manner to accomplish both the decrease in excitability and the restoration of synchrony seen in our biological data. In order to determine whether compensatory changes in $I_A$ occur in the intact network, we performed two-electrode voltage clamp in the same LC before and after 1 hr of TEA exposure. In order to track changes in $I_A$ in individual LCs we again used reversible sucrose block to perform voltage clamp immediately after acute TEA application and again after 1 hr of TEA perfusion. After voltage clamping anterior LCs, the sucrose was washed out and replaced with physiological saline, allowing the network to resume normal activity. This process was repeated after 1 hr of compensation.

## Biological methods: Data analysis

Intracellular burst waveforms were considered to begin with the first EPSP from pacemaker activity and ended upon return of the waveform to resting membrane potential. Recordings were analyzed using Clampfit 10.3 (Molecular Devices) and Spike 2 version 7 (CED, Cambridge, UK) software. Statistical analyses were performed using SigmaPlot 11.0. Correlation coefficients (R-values) were obtained by a Pearson correlation, and squared to calculate the coefficient of determination ($R^2$). Most data are 'before and after' effects within the same ganglion or cell, and therefore any two groups were compared with paired *t*-tests when the data were normally distributed, or Wilcoxon signed rank tests in the case of non-normality. The sample sizes for comparison of waveform

synchrony were calculated with power analyses based on projected means and standard deviations from data reported in our previous study with very similar experimental manipulations of TEA exposure of LCs (*Ransdell et al., 2013a*), which yielded target sample size of N=6–10 to yield a power of 0.8 to 0.97. Sample sizes for changes in network output and changes in $I_A$ after TEA exposure were based on similar data in our previous work (*Ransdell et al., 2012*), and yielded target sample sizes of N=5 to achieve a power of 0.909. Power analyses were conducted based on the use of paired *t*-tests to analyze the data. However, when data were not normally distributed we ended up using a Wilcoxon signed rank test, which was not utilized in our initial power analyses. All sample sizes used in our studies are reported in Figure Legends and/or in the Results section when significance values are reported.

To quantify synchrony of LC voltage waveforms, we performed a cross-correlation of the digitized voltage signal from LC pairs, as shown in *Figure 2B*. The first pacemaker spike was used to define the start of each LC burst, and bursts were considered to have terminated upon return to $V_{Rest}$ in the LCs. The coefficient of determination from this cross-correlation ($R^2$) was used to examine how accurately one burst waveform could predict the waveform in another LC (see *Ransdell et al., 2013a*). This cross-correlation was performed for every burst across the full time-course of the experiment (*Figure 2C*, *left*). A decrease in $R^2$ therefore indicates a loss or reduction in waveform synchrony. $R^2$ values from 10 consecutive bursts were averaged for all data points presented as waveform synchrony data, save for the individual points found in *Figure 2B and C*.

Five different measures of excitability of LCs were calculated using both extracellular and intracellular recordings (*Figure 2D*). Extracellular recordings from the ganglionic trunk were used to calculate the number of spikes per burst and spike frequency. Because our hypothesis predicts both increased LC excitability and desynchronization with TEA exposure, it should be noted that there is a potential confound in distinguishing these effects based on extracellular analysis alone. Axons of all LCs run through the ganglionic trunk, thus the increased spike count observed could result from an increase in the total number of action potentials, desynchronization of action potentials across LCs, or both. We therefore included three additional measures of excitability that helped clarify the effect. We measured the latency between the onset of SC pacemaker bursting to the first LC spike in each burst (*Figure 2D*; SC-LC phase delay). Two other measures of excitability were calculated from intracellular voltage changes: burst amplitude and total burst depolarization. Burst amplitude was defined as the maximal voltage change from $V_{Rest}$ to the highest peak of the burst, and total depolarization is the area under the curve above $V_{Rest}$, measured in mV*sec. For all measures of excitability, values from 10 consecutive bursts were averaged at each time-point in each preparation (N=8).

## Model methods: Single cell models

A detailed model was created with eight voltage-dependent conductances ($G_A$, $G_{Kd}$, $G_{NaP}$, $G_{CaS}$, $G_{CaT}$, $G_{CAN}$, $G_{SK(Ca)}$, $G_{BK(Ca)}$) and passive leak channels. These conductances are defined as: A-type potassium ($G_A$), delayed rectifier ($G_{Kd}$), persistent sodium ($G_{NaP}$), transient calcium ($G_{CaT}$), slow persistent calcium ($G_{CaS}$), calcium-dependent non-selective cation ($G_{CAN}$), two calcium-dependent potassium currents ($G_{SKKCa}$ and $G_{BKKCa}$), and leak ($G_{Leak}$). A single compartment model with biological dimensions for soma (*Ransdell et al., 2010*, *2013b*) was created in NEURON and its capacitance ($C_m$) and leak conductance ($G_{leak}$) were tuned to match the observed biological membrane time constant ($\tau$) and input resistance ($R_{in}$). This resulted in a soma with a length of 284.87 μm and a diameter of 125 μm, with a capacitance of 2.719 μF/cm$^2$. Channels were then added and their maximal conductances were tuned to match three biological properties observed, i.e., a) Total outward current b) Response to synaptic drive and c) Response to synaptic drive in the presence of TEA. These results were obtained from experiments performed on ligatured somata of *C. borealis* LCs (*Ransdell et al., 2012*, *2013a*, *2013b*). For the network studies another compartment termed Spike Initiation Zone (SIZ) was added to the model. This compartment was modelled as a cylinder with a capacitance of 1 μF/cm$^2$, a length of 400 μm, and a diameter of 8 μm, and contained only sodium, potassium and passive leak channels. This compartment allowed us to obtain spiking activity in the model cells for spike synchrony analysis. The resulting model equations were as follows:

$$C\frac{dV}{dt} = -I_A - I_{Kd} - I_{NaP} - I_{CaS} - I_{CaT} - I_{CAN} - I_{SKKCa} - I_{BKKCa} - I_{Leak} \quad \text{(Soma)}$$
$$C\frac{dV}{dt} = -I_{Na} - I_{Kdr} - I_{Leak} \quad \text{(SIZ)}$$

The individual currents were modelled as $I_c = g_{max,c} m^p h^q (V - E_c)$, where $g_{max,c}$ is its maximal conductance, $m$ its activation variable (with exponent $p$), $h$ its inactivation variable (with exponent $q$), and $E_c$ its reversal potential (a similar equation is used for the synaptic current but without $m$ and $h$). The kinetic equation for each of the gating functions $x$ ($m$ or $h$) takes the form

$$\frac{dx}{dt} = \frac{x_\infty(V, [Ca^{2+}]_i) - x}{\tau_x(V, [Ca^{2+}]_i)}$$

where $x_\infty$ is the steady state gating voltage- and/or $Ca^{2+}$- dependent gating variable and $\tau_x$ is the voltage- and/or $Ca^{2+}$- dependent time constant. The equations for the active channels in the soma compartment were fit using biological recordings for these currents from the cardiac ganglion of *Cancer borealis*. These currents were fit as follows: Voltage clamp data obtained with Clampfit were imported into MATLAB (Mathworks, Natick, MA) and fit using the MATLAB curve-fitting toolbox. Current data were converted to conductance data by dividing by $(V_m - E_{Rev})$, where $E_{Rev}$ was as follows: $E_{Na} = +55$ mV, $E_K = -80$ mV, $E_{Ca} = +45$ mV, $E_{Leak} = -50$ mV, and $E_{CAN} = -30$ mV. The time axis was adjusted to start from 0 for the beginning of the clamp. The following parameterization was used:

$$g(t) = \sum_{i=1}^{n} A_i \left(1 - \exp\left(\frac{-t}{\tau_{m,i}}\right)\right)\left(h_i - (h_i - 1)\exp\left(\frac{-t}{\tau_{h,i}}\right)\right)$$

In this equation, $A_i = G_{i,max} \times m_i$ was the maximal conductance of the current $i$ multiplied by its voltage-dependent steady-state activation ($m_i$), $h_i$ was the steady-state inactivation value, and $\tau_{m,i}$ and $\tau_{h,i}$ were the time constants with which activation and inactivation reached steady-state, respectively. This fitting procedure assumed that ionic currents were completely deactivated ($m = 0$) and de-inactivated ($h = 1$) prior to the onset of the voltage clamp. This was fit to each trace in a voltage clamp experiment, giving the values of each of the four parameters for each test clamp voltage ($V_c$). These values were then fit for each current as functions of $V_c$ using the general forms as stated below. This procedure yielded equations for the currents recorded in voltage clamp that could be used in simulations according to the Hodgkin-Huxley mathematical formalism.

$$A(V_c) = G_{max} \times m(V_c) = G_{max} \times \left(1 + \exp\left((V_c - V_{m,1/2})/k_m\right)\right)^{-1}$$
$$h(V_c) = \left(1 + \exp\left((V_c - V_{h,1/2})/k_h\right)\right)^{-1}$$
$$\tau_m(V_c) = \tau_{base,m} + \tau_{amp,m}\left(\exp\left((V_c - V_{\tau1,m})/k_{\tau1,m}\right) + \exp\left((V_c - V_{\tau2,m})/k_{\tau2,m}\right)\right)^{-1}$$
$$\tau_h(V_c) = \tau_{base,h} + \tau_{amp,h}\left(\exp\left((V_c - V_{\tau1,h})/k_{\tau1,h}\right) + \exp\left((V_c - V_{\tau2,h})/k_{\tau2,h}\right)\right)^{-1}$$

All the maximal conductances ($G_{i,max}$) were in $\mu S$, time constants in ms and voltages in mV.

## Model methods: Calcium dynamics

Intracellular calcium modulates the conductance of the calcium-activated potassium currents (BKKCa and SKKCa), calcium-activated nonselective cation current (CAN), and influences the magnitude of the inward calcium current in the LC (*Tazaki and Cooke, 1990*). A calcium pool was modelled in the LC with its concentration governed by the first-order dynamics (*Prinz et al., 2003*; *Soto-Treviño et al., 2005*) below:

$$\tau_{Ca}\frac{d[Ca^{2+}]}{dt} = -F \times I_{Ca} - \left([Ca^{2+}] - [Ca^{2+}]_{rest}\right)$$

where $F = 0.256\ \mu M/nA$ is the constant specifying the amount of calcium influx that results per unit (nanoampere) inward calcium current; $\tau_{Ca}$ represents the calcium removal rate from the pool; and $[Ca^{2+}]_{rest} = 0.5\ \mu M$. Voltage-clamp experiments of the calcium current (*Ransdell et al., 2013b*) showed the calcium buffering time constant to be around 690 ms ($\tau_{Ca}$).

**Table 1.** Nominal model conductance values:

| Conductance | Value (S/cm²) |
| --- | --- |
| Leak | 2e-4 |
| A | 6e-4 |
| BKKCa | 7.3e-3 |
| g1_Kd | 3e-4 |
| g2_Kd | 3.5e-5 |
| CaS | 1.7e-4 |
| CAN | 1.06e-4 |
| SKKCa | 8.79e-5 |
| CaT | 1.5e-4 |
| NaP | 3.06 e-4 |

## Model methods: Searching for LC neurons within the model parameter space

After creating a nominal LC model (*Tables 1,2*), we wanted to search the conductance space for other possible conductance combinations that might exhibit appropriate LC output. The properties that had to be maintained were; a) Input Resistance ($R_{in}$) and Resting Membrane Potential, b) Pre-TEA and Post TEA response to current injection c) Response to Synaptic drive obtained from biological cell.

The rules used to select the potential parameters were as follows (based on biological recordings): Synaptic Drive response should have an $R^2$ value of at least 0.8 or higher when compared to biological Synaptic Drive response. The duration of the pre-TEA response to a 6 nA, 50 ms current injection should be less than 120 ms. Also the peak should be less than −22 mV. The duration of the post-TEA ($G_{BKKCa}$ and $G_{Kd}$ reduced by 90%) response should be between 255–667 ms and its peak should be greater than −15 mV. A 9-D max conductance parameter space (5-fold variation over each conductance except $G_{Leak}$) was searched randomly for sets that satisfied the constraints above. We searched 20,000 different combinations of parameter sets with these criteria, and most of those which passed did not have a proper termination of activity following current injection (i.e., did not return to $V_{rest}$). We concluded this was due to an inappropriate relationship between $I_{CAN}$ and $I_{SKKCa}$. Subsequent trials revealed that a given ratio range (~1:0.83 respectively) of these two currents was necessary for proper termination of the activity. Larger ratios cause $V_{rest}$ to be higher due to the reversal potential (−30 mV) of CAN current. A higher fraction of $I_{SKKCa}$ (reversal potential −80mV) caused a large AHP after termination and reduced the duration of the post-TEA response. Using the updated selection criteria with a ratio $I_{CAN}$ to $I_{SKKCa}$, we found 180 parameter sets that passed. Of these 180 potential model sets, we selected only the ones that had Synaptic Drive response $R^2$ value > 0.9 compared to the biological Synaptic Drive response. This resulted in 49 potential parameter sets.

Biological data showed that $I_A$ and $I_{BKKCa}$ had a negative correlation in their magnitudes in LCs (*Ransdell et al., 2012*). We added this to our criteria for screening potential parameter sets for the network studies. We converted biological $I_A$-$I_{BKKCa}$ current data into factor data by dividing $I_A$ and $I_{BKKCa}$ by their respective factor average. $G_A$ and $G_{BKKCa}$ values of passed parameters were similarly divided by its average to get its factor data. The biological data were fitted using a linear polynomial from 95% to 70% confidence intervals, in steps of 10%. For network studies we used a 70% confidence interval, which left us with 14 potential parameter sets that represented LC model neurons for use in modeling studies.

## Model methods: Development and validation of a population of conductance-based model networks for studying mechanisms restoring network synchrony

Our results demonstrate that intact cardiac ganglia are able to compensate for the loss of high-threshold K⁺ currents and restore both excitability and synchrony within one hour of TEA blockade. We next set out to explore the mechanisms by which excitability and synchrony could be restored in this network. To maximize our ability to interrogate multiple parameters that may be responsible for compensation in this system, we constructed a population of conductance-based biophysical models of the CG network. This allowed us to simulate the TEA conductance blockade and then manipulate individual conductances, both voltage-gated and synaptic, to examine their effects on network excitability and synchrony.

Our 14 parameter sets for LCs were used to create 50 random 5-cell networks of LCs, ensuring that the same model LC never appeared twice in the same network. The five cells within a network

**Table 2.** Model current parameters.

| $I_{ion}$ | $x^P$ | $x_\infty$ | $\tau_x$ (msec) |
|---|---|---|---|
| $I_A$ | $m^3$ | $\frac{1}{1+\exp((V+21.46)/-17.96)}$ | $3.002 + \frac{4.073}{1+\exp((V+24.18)/2.592)}$ |
| | h | $\frac{1}{1+\exp((V+21.14)/25.99)}$ | $9.434 + \frac{11.7}{1+\exp((V+1)/5.317)}$ |
| $I_{CaS}$ | $m^2$ | $\frac{1}{1+\exp((V+24.75)/-5)}$ | $20 + \frac{50.2}{\exp((V+20.25)/1)}$ |
| | h | $\frac{45}{40+[Ca^{2+}]}$ | $\frac{1}{0.02}$ |
| $I_{CaT}$ | m | $\frac{1}{1+\exp((V+20)/-1.898)}$ | $18.51 - \frac{3.388}{\exp((V-6.53)/9.736)+\exp((V+12.39)/-2.525)}$ |
| | h | $\frac{1}{1+\exp((V+55.27)/6.11)}$ | $20.23 + \frac{40}{\exp((V+23.48)/-9.976)+\exp((V+5.196)/10.84)}$ |
| $I_{kd}$ | $m_1{}^4$ | $\frac{1}{1+\exp((V+24.19)/-10.77)}$ | $25.049 + \frac{25}{1+\exp((V+25.84)/6.252)}$ |
| | $h_1$ | $0.3 + \frac{1-0.3}{1+\exp((V+15.87)/5.916)}$ | $550 + \frac{954.9}{\exp((V+10.8)/-15)}$ |
| | $m_2{}^4$ | $\frac{1}{1+\exp((V+23.32)/-10)}$ | $100 + \frac{550}{\exp((V+15)/12.46)}$ |
| $I_{NaP}$ | $m^3$ | $\frac{1}{1+\exp((V+32.7)/-18.81))}$ | $3.15 + \frac{0.8464}{\exp((V+0.8703)/-6.106))}$ |
| $I_{CAN}$ | w | $(0.0002*[Ca^{2+}]\wedge2/(0.0002*[Ca^{2+}]\wedge2+0.05))$ | $(40/(0.0002*[Ca^{2+}]\wedge2+0.05))$ |
| $I_{SKKCa}$ | w | $(0.0001*[Ca^{2+}]\wedge2/(0.0001*[Ca^{2+}]\wedge2+0.1))$ | $(4/(0.0001*[Ca^{2+}]\wedge2+0.1))$ |
| $I_{BKKCa}$ | a | $\frac{[Ca^{2+}]}{1+\exp((V-15+0.08*[Ca^{2+}])/-15)*(1+\exp((V+5+0.08*[Ca^{2+}])/-9)*(2+[Ca^{2+}]}$ | $\frac{1}{0.4}$ |
| | b | $\frac{7}{5+[Ca^{2+}]}$ | $\frac{1}{0.2}$ |

F = Faradays constant

R = Gas constant

V = Membrane voltage

$[Ca^{2+}]$ = Calcium concentration

were then electrically coupled using conductance values tuned to reflect experimental observations of coupling coefficients. Small cell (SC) pacemaker drive was simulated as excitatory synapses via the NetStim function in NEURON. Parameters for the model of the synaptic drive onto LCs were tuned to get 6 to 9 spikes in the nominal LC model. It was observed biologically that frequency of SC firing increases within the slow wave oscillation cycle of LCs. Based on these recordings, the model SC burst initially fired at 18 Hz for first 440 ms and then increased to 25 Hz for 560 ms, with the burst terminating at 1000 ms.

Our experimental TEA block was simulated in these networks by reducing $G_{BKKCa}$ and $G_{Kd}$ conductances by 90% in the 3 anterior LCs based on biological data in LCs (*Ransdell et al., 2013a*). We imposed a final set of selection criteria on the randomly generated model networks, rejecting networks that increased synchrony or decreased the total number of spikes after the simulated TEA block, as this was never observed in biological networks. This left 27 networks that reproduced the biological trends and these were used in subsequent analyses to explore potential conductance changes that could restore network synchrony.

Somatic burst potentials drive action potentials in LC axons, so divergent burst waveforms would be expected to cause desynchronized spiking. Our biological data qualitatively agreed with this, but a precise quantification of synchrony for all spikes within a burst is subject to many ambiguities. Our model networks easily provided precise spike times for each cell in the network, so we chose to examine actual spike synchrony in the model to complement the burst waveform analysis in the biological preparation. Our analysis considered synchrony for paired anterior LCs with a nominal coupling conductance of 0.0182 S using a 25 ms bin width for spike-times (*Wang and Buzsáki, 1996*). Spikes occurring in both cells during the same bin were considered synchronized, while spikes that did not bin together were tallied as desynchronized. Using the definition of synchrony listed in the next section, these randomly generated model networks exhibited 'control' synchrony scores ranging from 0.642 to 1.0 with a median value of 0.915 (matching data in biology), where 1.0 represents perfect spike synchrony.

## Model methods: Data analysis

In the models, spike synchrony between two cells was calculated based on spike times (*Wang and Buzsáki, 1996*). Spike times were recorded from each LC's Spike Initiation Zone (SIZ). The simulation time was divided into 25 ms bins. After initializing all bins to zero, each cell spike was added to the corresponding bin. Synchrony (SY) between two cells A and B was calculated using the following equation:

$$SY_{AB} = \frac{\sum_{l=1}^{k} A(l)*B(l)}{\sqrt{\sum_{l=1}^{k} A(l) * \sum_{l=1}^{k} B(l)}}$$

where $l$ is the current bin and $k$ is the maximum number of bins. Spikes occurring in both cells during the same bin were considered synchronized, while spikes that did not bin together were tallied as desynchronized.

To compare the measures of spike and waveform synchrony in model networks, model waveform synchrony measures (*Figure 3*) were performed as described above on filtered voltage traces (Gaussian lowpass filter, 15 Hz cutoff frequency; Clampfit 10.3) that remove the influence of the axonal spikes on the voltage waveforms. Statistical analyses were performed in SigmaPlot v11.0. The effects of changing $G_{max}$ on the number of spikes and synchrony among TEA-'treated' model neurons were tested with paired $t$-tests. Analyses changes in spike number and spike synchrony with changing coupling and synaptic strengths (*Figure 5*) were analyzed with One-Way ANOVAs with post-hoc pairwise comparisons between a given percent change and the TEA case conducted via Holm-Sidak tests.

## Acknowledgements

This work was supported by NIH grant MH46742 (DJS and E Marder), NIH grant MH087755 (SSN), NIH grant NIGMS 5T32GM008396 (BJL), NSF grant CNS-1429294 (SSN), and a grant from the University of Missouri Research Board (DJS and SSN).

## Additional information

### Funding

| Funder | Grant reference number | Author |
| --- | --- | --- |
| National Institute of General Medical Sciences | 5T32GM008396 | Brian J Lane |
| University of Missouri | Research Board | Satish S Nair<br>David J Schulz |
| National Institutes of Health | MH087755 | Satish S Nair |
| National Science Foundation | CNS-1429294 | Satish S Nair |
| National Institutes of Health | MH46742 | David J Schulz |

The funders had no role in study design, data collection and interpretation, or the decision to submit the work for publication.

### Author contributions

BJL, PS, Conception and design, Acquisition of data, Analysis and interpretation of data, Drafting or revising the article; JLR, Conception and design, Acquisition of data, Analysis and interpretation of data; SSN, DJS, Conception and design, Analysis and interpretation of data, Drafting or revising the article

### Author ORCIDs

David J Schulz, http://orcid.org/0000-0003-4532-5362

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
