## [Decision Letter]

Thank you for submitting your article "Synergistic Plasticity of Intrinsic Conductances and Electrical Synapses Restores Synchrony in an Intact Motor Network" for consideration by *eLife*. Your article has been reviewed by three peer reviewers, one of whom, Ronald L Calabrese, is a member of our Board of Reviewing Editors and the evaluation has been overseen by a Senior Editor. The following individuals involved in review of your submission have agreed to reveal their identity: Carlos D Aizenman (Reviewer #2) and Farzan Nadim (Reviewer #3).

The reviewers have discussed the reviews with one another and the Reviewing Editor has drafted this decision to help you prepare a revised submission.

Summary:

In this innovative study, the authors use a combination of pharmacology, electrophysiology, and modeling approaches in an intact neuronal network (crab cardiac ganglion, CG, in vitro) to uncover a synergistic plasticity of an intrinsic conductance and electrical coupling in the recovery normal synchronous activity following pharmacological challenge.

In the intact CG network, blocking a subset of K conductances in normally synchronous motor neurons (LCs) with TEA resulted in both hyperexcitability and loss of synchronized activity. Over the course of 30-60 minutes synchronization returns and hyperexcitability moderates: compensation. Using an ensemble modeling approach to guide experimentation, the authors identified two distinct and potentially synergistic mechanisms by which motor neurons could compensate and recover synchronous activity: plasticity of an intrinsic voltage-gated conductance (A-type K^+^ current, IA) and plasticity of LC electrical coupling. Intrinsic compensation alone in the models was insufficient to fully restore synchrony across motor neurons, but a concomitant increase in electrical coupling ensured nearly complete resynchronization. Modeling also suggests that although a sufficient increase in electrical coupling alone could restore full synchrony, it could not simultaneously restore the original level of excitability. Direct measurements verify both an increase in IA and an increase in electrical coupling conductance associate with re-synchronization in the TEA challenged intact CG over the course of 30-60 minutes. The authors conclude that multi-component mechanisms are likely necessary for full compensation, and that their synergistic action is potentially more efficient than either mechanism operating in isolation.

Essential revisions:

1) The models are not adequately described so their 'predictions' seem less compelling. What exactly does this model look like? Is it really representative of these neurons in any reasonable way? Please show traces of the voltage time series (potentially also the ongoing IA and the gap-junctional current) for a representative example. The fact that IA compensates for other K currents is not too surprising and rather intuitive. So is the fact that an increase in electrical coupling can increase synchrony. So what is the model saying that is not intuitively obvious? The model needs to be properly analyzed and described or removed. Is the point to show that other currents cannot account for the compensation; if so does the models really show this or does it show that in isolation their modulation cannot do both a decrease in excitability and an increase in synchrony? More analysis and explanation of the modeling is needed.

2) Although the findings in this study are extremely interesting, the premise on which the study is introduced does not provide the accurate context for these results. The authors have insisted on equating neuromodulation with destabilizing perturbations. Yet modulators, under biological conditions, do not destabilize insomuch as they define a new stable state for the network and, as such, they fit better into the framework of flexibility. Although there are examples in which long-term modulation leads to compensatory feedback (such as in the work of the Baro lab on tonic dopamine modulation), more often modulators simply modify the network output. It would be best if the authors modify the Introduction to simply reflect how cellular and network level compensatory mechanisms can provide robustness in response to extrinsic perturbations (such as injury perhaps), and remove the description of how neuromodulators may act in face of ionic current variability. In the same vein, the results presented on the effect of serotonin on the LCs seem to be preliminary and the effects are not properly described or analyzed (e.g. no raw traces, no dose-response analysis to establish physiological concentrations). It is unclear if the selective application site is appropriate, if there is a cellular level compensation mechanism at play, or if it is related to the TEA results. As such, it would be best if the authors remove the serotonin figure and data. If the serotonin results can be developed further and properly contextualized they could be submitted in future as an Advance at *eLife*.

3) The really novel aspect of the study is that gap-junctional conductance can participate in homeostatic regulation on network activity. This point could be better emphasized while the relevance to neuromodulation is deemphasized/removed.

---

## [Author Response]

*Essential revisions:*

*1) The models are not adequately described so their 'predictions' seem less compelling. What exactly does this model look like? Is it really representative of these neurons in any reasonable way? Please show traces of the voltage time series (potentially also the ongoing IA and the gap-junctional current) for a representative example. The fact that IA compensates for other K currents is not too surprising and rather intuitive. So is the fact that an increase in electrical coupling can increase synchrony. So what is the model saying that is not intuitively obvious? The model needs to be properly analyzed and described or removed. Is the point to show that other currents cannot account for the compensation; if so does the models really show this or does it show that in isolation their modulation cannot do both a decrease in excitability and an increase in synchrony? More analysis and explanation of the modeling is needed.*

We understand that the intent and contributions of the model may not have been as transparent to the reviewers as it should have been. The revised version has corrected for this as noted below, including a thorough revamp of Figure 3, Figure 5 and Figure 7. The model is indeed not intended to represent a comprehensive examination of the parameter space of these cells and networks. Yet the modeling approach was valuable in framing our experimental design, and we think provides impact in the following ways, and we have attempted to add text throughout the manuscript to support these assertions:

1) The model is able to address whether the calculation of synchrony based on LC membrane potential waveform and the calculation using LC spikes had functional overlap. Because the biological system does not allow for straightforward recording of spiking in multiple individual neurons, the model allowed us to translate waveform synchrony into a measure of spike synchrony to assess these questions at the ultimate level of output – spiking. The model showed that waveform synchrony correlated very well with spike synchrony in all cases.

2) While it is not difficult to intuit that an overall suite of changes to underlying conductance could restore output, it actually was not clear or intuitive to us as to whether the maximal change in conductances of ‘only’ the A current that we found in biology (Ransdell et al. 2012) would be sufficient in isolation to have a profound impact on network level properties of output (i.e., synchrony). The model was able to provide a prediction in this case, as well as shed interpretive light that perhaps widespread reorganization of conductances is not necessary for such potent compensatory processes.

3) Finally, the model helped address whether any other current (CaS, CaT, SKKCa, CAN, Nap), might similarly be able to restore excitability and synchrony on its own. We showed this is not the case at the single current level, by perturbing conductances of all currents individually. In addition to perturbing only individual conductances, we also varied current kinetics and activation parameters (half-activation voltage V1/2, +/- 10 mV, and slope factor k, by 0.5 and 2 (Ballo et al., 2012) and time constant by +/10 ms) for all the cell currents individually, and found that no changes in parameters for a single current could simultaneously restore excitability at the single cell level, and synchrony at the network level (data not shown). While the analysis has focused only on the parameters of a single current, simultaneous changes in parameters of multiple currents could also potentially provide similar compensation, and that remains to be explored.

4) The synergistic nature of the combined A and coupling changes were only able to be thoroughly conducted via computational approaches, and this is a focal point of interpretation for the entire study.

The model thus adds predictive and interpretive value. We hope the above justification and changes made in the manuscript help make this case.

*2) Although the findings in this study are extremely interesting, the premise on which the study is introduced does not provide the accurate context for these results. The authors have insisted on equating neuromodulation with destabilizing perturbations. Yet modulators, under biological conditions, do not destabilize insomuch as they define a new stable state for the network and, as such, they fit better into the framework of flexibility. Although there are examples in which long-term modulation leads to compensatory feedback (such as in the work of the Baro lab on tonic dopamine modulation), more often modulators simply modify the network output. It would be best if the authors modify the Introduction to simply reflect how cellular and network level compensatory mechanisms can provide robustness in response to extrinsic perturbations (such as injury perhaps), and remove the description of how neuromodulators may act in face of ionic current variability. In the same vein, the results presented on the effect of serotonin on the LCs seem to be preliminary and the effects are not properly described or analyzed (e.g. no raw traces, no dose-response analysis to establish physiological concentrations). It is unclear if the selective application site is appropriate, if there is a cellular level compensation mechanism at play, or if it is related to the TEA results. As such, it would be best if the authors remove the serotonin figure and data. If the serotonin results can be developed further and properly contextualized they could be submitted in future as an Advance at eLife.*

Thank you for this comment. We understand and appreciate the spirit of the comment and have removed the serotonin modulation from the manuscript, to the betterment of clarity for the study.

*3) The really novel aspect of the study is that gap-junctional conductance can participate in homeostatic regulation on network activity. This point could be better emphasized while the relevance to neuromodulation is deemphasized/removed.*

Agreed. We have re-written the manuscript with an attempt to change the emphasis and clarify the goals and outcomes of the study. Hopefully the reviewers feel we’ve accomplished this change in emphasis.